# Factors Influencing Performance of Exercise Behavior of Middle-Aged Men with Chronic Disease Based on the Information–Motivation–Behavioral Skill-Revealed-Related Variables (IMBR) Model

**DOI:** 10.3390/healthcare11010100

**Published:** 2022-12-28

**Authors:** Hee-Kyung Kim, Hyoungtae Kim, Jeong-Hyo Seo

**Affiliations:** 1Department of Nursing, Kongju National University, Gongju 32588, Republic of Korea; 2Department of Convergence Management, Corporate Management Major Endicott, College of International Studies, Woosong University, Daejeon 34606, Republic of Korea; 3Department of Nursing, Graduate School, Kongju National University, Gongju 32588, Republic of Korea

**Keywords:** exercise, information, motivation, perceived barriers to exercise, exercise self-efficacy, hierarchical multiple regression, Information–Motivation–Behavioral skill model

## Abstract

The purpose of this study was to analyze the factors influencing performance of exercise behavior of middle-aged men with chronic disease by adding variables through literature to the Information–Motivation–Behavioral skill model. Subjects of this study are total 171 people belonging to exercise clubs. In the results of putting the control variable in the step-1 of the hierarchical regression analysis, the health condition, smoking, the number of exercises per week, and hours of each exercise were revealed as influence factors and showed 38.4% explanatory power on the performance of exercise behavior. In the results of putting the factors required for behavioral change in the step-2 analysis, the information for exercise, motivation for exercise, sport commitment, and perceived barriers to exercise were influence factors, showing 60.1% explanatory power on the performance of exercise behavior. In the results of putting the exercise self-efficacy of exercise behavioral skills in the step-3 analysis, it was revealed as an influence factor that showed 63.0% explanatory power. Regarding the influence on participants’ exercise behavior, the factors required for behavioral change and behavioral skill factors were relatively more important than the general characteristics. This study suggests application of IMBR model to the program for exercise behavior.

## 1. Introduction

Owing to the improvement of national income level and development of medical technology in Korea, the life expectancy and health span are continuously increasing while the percentage of people who regard themselves as healthy is ranked the lowest among the Organization for Economic Cooperation and Development (OECD) countries [1]. After realizing the great importance of national disease prevention and health promotion in 2022, the state is continuously investing budget in the reinforcement of manpower, which is not sufficient yet [1]. Meanwhile, with the rise of new types of health demands and supplies, there has been increased interest in health maintenance and enhancement, longevity, and happy life, while people’s demands for health care are also changing from cure to wellness [1,2]. Moreover, as the importance of physical activity and exercise is rising in the level of preventing chronic diseases such as obesity, high blood pressure, and diabetes, the state is choosing physical activity including exercise as one of the main tasks for the practice of healthy life [1]. Regardless of age or gender, the most important element to maintain and improve health is physical activity and exercise, and people who regularly exercise or naturally perform physical activity in daily life can feel their own mental and physical health more positive, enhance their own health condition, and also protect their functional independence in the aging process [2].

In the results of surveying the athletic population, unfortunately, only 23% of the US adult population exercised enough to have effects on health, and 36.2% of adults did not perform physical activity at all during their leisure time [2]. According to the results of the Korea National Health and Nutrition Examination Survey in each gender and lifecycle, it would be necessary to pay attention to middle-aged men. Only 42% of middle-aged men performed physical activity; their practice rate of aerobic physical activity was much lower than the one of youth population; and the practice rate of aerobic physical activity was gradually declined as they got older [3]. Especially, the middle age is in the physical, psychological, and social transition period. The middle-aged men start experiencing the decline of male hormones, decrease in physical ability, distinct aging phenomenon, decelerated metabolism, decrease in cardiac function and respiratory activity, gradually weakened functions of frame and muscle, increased risk of adult diseases such as arthritis, diabetes, heart disease, and high blood pressure, declined confidence and depression, and accompanied decline of mental function such as social phobia, so it is important not to neglect health care [4].

Especially, the physical change and symptoms of disease are more slowly shown in men than women. For this reason, most of them neglect to actively respond to them, so it would be necessary to induce them to regularly perform physical activity and exercise, which is one of health care. Furthermore, if the middle-aged men constantly perform exercise such as aerobic exercise, stretching, and muscle strengthening exercise of health guidelines, they can not only see the effects on the improvement of muscle strength, increase in bone mineral density, maintenance of blood sugar, and decrease in blood fat [4], but also positively lead their middle age and enjoy the preparation for old age [4,5].

Meanwhile, the Information–Motivation–Behavioral skill model is a theory that has conceptualized the influence factors for starting and continuing the health behavior as information, motivation, and behavioral skill factors. If a subject fully acquires information related to behavior, the subject is motivated for the behavior. Once the behavioral skill is improved, the behavioral change and maintenance is accelerated, so the objective and subjective health conditions are enhanced [6]. In the results of research for changes in exercise behavior and performance enhancement targeting chronically ill patients [7,8], the exercise-related health behavior could be well performed only when they had the provision of information related to effective exercise, motivation for exercise behavior, and behavioral skill related to effective exercise, that is, exercise self-efficacy.

Information is individual’s knowledge about behavioral change [6]. When there is more knowledge about physical activity is more, the amount of physical activity could be increased [9], so the knowledge and information for exercise are predictive factors of exercise behavior [10]. As an internal process that starts a behavior when pursuing a new goal, the motivation for participating in exercise is the power to start, activate, and continue the behavior [11]. Moreover, as a desire to achieve a goal regarded as valuable, it has important effects on the performance of exercise behavior [5,12]. According to the research by Nam [5], the motivation for participation and sport commitment had effects on the intention to continuously participate in exercise targeting the middle-aged men of foot volleyball club. Additionally, in the research [13] targeting the middle-aged male workers, the autonomous motivation had effects on the stage of changes in exercise behavior. Additionally, the motivation for exercise shows a positive correlation with exercise self-efficacy [14], and it performs health behavior, that is exercise behavior through behavioral skill, that is exercise self-efficacy.

The behavioral skill is a subjective belief in the ability to perform behavioral change, which is called exercise self-efficacy [6,7,8]. In case of middle-aged men, when the stage of change in exercise got higher, the exercise self-efficacy got higher [15]. Additionally, the exercise self-efficacy was a main factor affecting the performance of exercise behavior [16]. Once the exercise self-efficacy, that is the confidence in the ability to overcome any tasks or barriers is raised, it is possible to maintain exercise well [17]. Therefore, the Information–Motivation–Behavioral skill model that is handling the concept of health-related behavior could be an important theory for applying a self-management program [18] or predicting and explaining middle-aged men’s performance of exercise behavior, so it was planned to preferentially apply the Information–Motivation–Behavioral skill model to this study.

However, the Information–Motivation–Behavioral skill model was not good enough to explain the adults’ performance of exercise behavior [7,19], so the researchers added the relevant variables affecting the performance of exercise behavior. Thus, in the results of evaluating the factors related to adults’ performance of exercise behavior after middle age, which had been sporadically demonstrated in preceding researches, this study added sport commitment [5,12], perceived benefits of exercise and perceived barriers to exercise [20,21,22], and exercise-related social support [20] as the factors required for behavioral change, named them “revealed related-variables”, and aimed to use the IMBR model as the conceptual framework.

Sport commitment refers to the optimal psychological state that occurs when you are completely immersed in exercise activities [12]. As the sport commitment is highly related to the performance of exercise behavior, when the state of sport commitment is high, ordinary people can feel joy and accomplishment of sport and also experience the climax without any interruption [23,24], which becomes the source of motivation for physical activity [12]. The higher the degree of sport commitment of middle-aged men, the higher the intention to continue participation [5] and the higher the performance of exercise behavior. The exercise self-efficacy and perceived benefits of exercise had much greater effects on the degree of performance of exercise behavior in the adults who regularly performed exercise behavior than the people who did not regularly exercise. As the perceived barriers to exercise had much less effects on it [21], they were revealed as main factors related to the performance of exercise behavior. Additionally, the perceived benefits of exercise had effects on the performance and level of exercise behavior of middle-aged adults and middle-aged women. The higher the benefit of exercise and the lower the disability of exercise, the higher the ability to perform exercise [22]. Middle-aged men will recognize the benefit of exercise and the lower the disability of exercise, the better they will perform exercise behavior. When they received exercise-related social support, the exercise behavior was performed well [20]. Thus, the cognitive and emotional variables such as sport commitment, perceived benefits of exercise, perceived barriers to exercise, and exercise-related social support were added as the factors required for behavioral change of Information–Motivation–Behavioral skill model.

Therefore, the purpose of this study is to provide the basic data for the development of programs for enhancing the performance of exercise behavior by applying the IMBR Model to middle-aged men, conducting the correlation analysis on information for exercise, motivation for exercise, sport commitment, perceived benefits of exercise, perceived barriers to exercise, exercise-related social support, exercise self-efficacy, and performance of exercise behavior, and revealing the factors affecting the performance of exercise behavior through the hierarchical regression analysis.

## 2. Conceptual Framework of This Study

Figure 1 is the conceptual framework of this study. First, the Information-Motivation, Behavioral Skill, and Revealed-related Exercise (IMBR) model was composed by adding the factors required for changes in exercise behavior revealed through literature review such as sport commitment, perceived benefits of exercise, perceived barriers to exercise, and exercise-related social support to the information, motivation, behavioral skill, and behavioral result, presented in the Information-Motivation and Behavioral Skill Model by Fisher [6].

## 3. Materials and Methods

### 3.1. Participants

As the research subjects, this study randomly sampled the people who were exercising at ordinary times as the members of exercise clubs in each S1 city, S2 city, and G city, among the middle-aged men in their 41–59 residing in communities. Following the preceding research [15] to test influences between variables through the hierarchical multiple regression analysis by using the G-power 3.1.9.7. Program, the number of samples required for maintaining 14 predictive variables, 0.15 effect size, 0.05 significance level, and 0.90 test power was calculated as 171 people. Considering the 5% dropout rate, including the number of unavailable questionnaires, such as collection difficulties and unavailable questionnaires, it was calculated as 179. In the results of distributing and collecting 179 questionnaires, total 171 questionnaires were used for the final analysis after excluding eight questionnaires that were out of the relevant age.

Criteria of selection: As the members of exercise club in their 41–59, residing in communities, exercising in their daily lives, understood the purpose of research and how to collect data, and allowed to participate in the study.

Criteria of exclusion: The middle-aged men who are suffering from acute diseases or registered as the disabled

### 3.2. Procedures

In this study, the data were collected from 24 October to 1 November 2022 after getting review on research purpose, methods, subjects, guarantee of rights, and questionnaire from the K University Institutional Review Board. Before data collection, the researchers got permission for data collection after explaining the purpose and content of the research by visiting the presidents of each exercise club in S1 city, S2 city, and G city. Obtaining the gathering schedule of exercise clubs, the researchers visited the places of gathering on the scheduled dates. After explaining the research purpose and how to fill out the questionnaire to total 179 research subjects including 100 subjects in S1 city, 40 subjects in S2 city, and 39 subjects in G city, each individual was told to read the research explanation, to sign the consent form, and then to fill out the questionnaire. It took about 20 min to fill out the questionnaire. In the future, the results will be feedback to the subjects to increase the meaning and significance of participation in the study.

### 3.3. Measures

The instrument used for this study is self-administered questionnaire composed of 14 items about general characteristics, 14 items about information for exercise, 13 items about motivation for exercise, 12 items about sport commitment, 18 items about perceived benefits of exercise, 10 items about perceived barriers to exercise, seven items about exercise-related social support, 13 items about exercise self-efficacy, and seven items about performance of exercise behavior.

#### 3.3.1. Information for Exercise

The researchers completed the instrument by referring to the literature of Kim et al. [2] and the contents of information for exercise presented in Song’s [25] study. After that, two exercise experts and one professor of physical education checked the validity of the instrument. It has total 14 items composed of eight items about understanding of exercise and six items about effects of exercise. Each item can be responded as ‘Yes’, ‘No’, or ‘I do not know’. When getting the correct answer, 1 point was applied. When providing the wrong answer or responding as “I do not know”, 0 point was applied. The mean score is in 0–1 point. The higher mean score means the correct understanding of exercise-related information. The reliability Cronbach’s ⍺ was 0.89.

#### 3.3.2. Motivation for Exercise

The motivation for exercise is composed of personal motivation and social motivation. To measure the motivation for exercise, this study used the instrument modified by Jin [26] based on the instrument used by Karvinen et al. [27] who applied Ajzen’s Theory of Planned Behavior to exercise behavior. The personal motivation for exercise is an attitude toward exercise, composed of total seven items including three items about emotional attitude and four items about instrumental attitude. Additionally, the social motivation for exercise is subjective norms of exercise, composed of total six items including three items about injunctive norms and three items about descriptive norms. Each item is measured on the basis of Likert 7-point scale: 1 point is described with negative verb while 7 point is described with positive verb. The higher mean score means the higher degree of motivation for exercise. In this study, the Cronbach’s ⍺ for the reliability of the personal motivation was 0.94; the social motivation was 0.84; and the total reliability was 0.93.

#### 3.3.3. Sport Commitment

This study used the instrument for sport commitment developed by Jeong [28] was used. It is composed of total 12 items including eight items about cognitive commitment and four items about behavioral commitment. Each item is measured on the basis of Likert 5-point scale (1point for ‘Hardly ever’, 5points for ‘Very much so’). The higher mean score means the higher degree of sport commitment. In this study, the Cronbach’s ⍺ for the reliability was 0.77.

#### 3.3.4. Perceived Benefits of Exercise

This study used the instrument developed by Lee [22] based on the instrument by Sechrist et al. [29] and the instrument by Stenhardt, Dishman [30]. This instrument is on the basis of Likert 5-point scale (1 point for ‘Hardly ever’, 5 points for ‘Very much so’). The higher mean score means the highly perceived benefits of exercise. In this study, the Cronbach’s ⍺ for the reliability was 0.92.

#### 3.3.5. Perceived Barriers to Exercise

This study used instrument that Oh [20] modified instrument developed by Lee [22] based on instrument by Sebrist et al. [29] and instrument by Stenhardt, Dishman [30]. Composed of total ten items, each item is measured on the basis of Likert 5-point scale (1 point for ‘Hardly ever’, 5 points for ‘Very much so’). The higher mean score means the highly perceived barriers to exercise. In this study, the Cronbach’s ⍺ for the reliability was 0.86.

#### 3.3.6. Exercise-Related Social Support

The exercise-related social support was measured by using the instrument modified by Choi [31] based on the instrument developed by Sallis et al. [32] and the social support instrument by Park [33]. In relation to exercise, there are total seven items including two items about emotional support, two items about informative support, two items about material support, and one item about appraisal support, and each item is measured on the basis of Likert 5-point scale (1 point for ‘Not at all’, 5 points for ‘Very much so’). The higher mean score means the high degree of exercise-related social support. In this study, the Cronbach’s ⍺ for the reliability was 0.78.

#### 3.3.7. Exercise Self-Efficacy

The exercise self-efficacy was measured by using the instrument modified by Choi [31] based on the instrument by Dzewaltowski [34] and the tool by Sallis et al. [32]. The instrument is composed of total 13 items including three items about self-efficacy related to ‘exercise types or methods’, five items about self-efficacy of ‘making time for exercise’, and five items about self-efficacy of ‘performing exercise even with barriers’. The degree of feeling confidence in successfully performing exercise was measured from 0 point to 10 points in each item. The higher mean score means the high self-efficacy of exercise. In this study, the Cronbach’s ⍺ for the reliability was 0.92.

#### 3.3.8. Performance of Exercise Behavior

The degree of performance of exercise behavior was measured by using the “exercise behavior performance tool I” written by Lee [22] based on the health promotion lifestyle tool by Walker et al. [35] and the tool by Park [36]. It is composed of total seven items on the basis of Likert 4-point scale (1 point for ‘Hardly ever’, 4 points for ‘Always do’). The higher mean score means the higher degree of performance of exercise behavior. In this study, the Cronbach’s ⍺ for the reliability was 0.85.

### 3.4. Statistical Analysis

Using the SPSS Window 25.0 Program, the general characteristics and degree of variables were calculated through the arithmetic statistics such as percentage. To compare the degree of performance of exercise behavior, this study conducted the t-test and ANOVA, and also used the Scheffe test for the post-analysis. The correlations of information for exercise, motivation for exercise, sport commitment, perceived benefits of exercise, perceived barriers to exercise, exercise-related social support, exercise self-efficacy, and performance of exercise behavior of research subjects were analyzed through Pearson’s correlational coefficients. The factors affecting the performance of exercise behavior of research subjects were analyzed by using the hierarchical multiple regression analysis.

### 3.5. Ethical Consideration

This study was approved by the K University’s Institutional Review Board for the purpose, methodology, and protection of the rights of participants (KNU_IRB_2022-116). The guidelines for ethical research were followed during the study period. The consent form contained information on anonymity and confidentiality and explained that even after consenting to the research participation according to the person’s voluntary intention, he/she could stop participating in the research at any time and there was no disadvantage there from. The collected information will be managed in accordance with the Personal Information Protection Act, and the author will do her best to ensure the confidentiality of all personal information obtained through research. Informed consent was obtained from all subjects involved in the study. It was informed that the collected data will be stored for 3 years in a lockable cabinet accessible only by the author and will be discarded using a shredder after statistical analysis by computational coding for subject anonymity.

## 4. Results

### 4.1. General Characteristics of Participants

Table 1 shows the general characteristics of participants. The total number of middle-aged male adults was 171. The range of age was 41–59 years old. The average age was 47.12 ± 4.57 years, and 66.1% (113 persons) aged 41–49 accounted for the majority. The participants with no religion accounted for the majority (63.7%, 109 persons). Most of the participants were married (92.4%, 158 persons) for marital state, showed the middle or higher economic level (86.5%, 148 persons), and graduated from college or higher (85.4%, 146 persons).

Over the majority was office workers (59.6%, 102 persons) for job, and also positively perceived their health state (67.3%, 115 persons). Most of the participants did not smoke (77.2%, 132 persons) while the participants who responded ‘Yes’ to drinking accounted for over the majority (50.1%, 105 persons). In the number of diseases, most of the participants (96.5%, 165 persons) had one or more. Over the majority was exercising four times or less per week (74.9%, 127 persons) for the number of exercises per week, and also exercising for less than one hour (71.9%, 123 persons) for hours of each exercise.

### 4.2. Comparison of Differences in the Performance of Exercise Behavior According to the General Characteristics of Middle-Aged Men

Table 1 is the results of differences in the performance of exercise behavior according to the general characteristics of middle-aged men. The men in their 50–59 showed the higher degree of performance of exercise behavior than the men in their 41–49 (t = −2.09, *p* = 0.038) while the subjects with no religion showed the higher degree of performance of exercise behavior than the subjects with religion (t = 2.09, *p* = 0.038). The subjects who perceived their economic level as ‘high’ showed the higher degree of performance of exercise behavior than the subjects who perceived it as ‘middle’ and ‘low’ (F = 7.53, *p* = 0.001) while the subjects who perceived their health condition as good showed the higher degree of performance of exercise behavior than the subjects who perceived it as bad (t = −7.67, *p* < 0.001).

The subjects who did not smoke showed the higher degree of performance of exercise behavior than the subjects who smoked (t = 4.35, *p* < 0.001) while the subjects who did not drink showed the higher degree of performance of exercise behavior than the subjects who drank (t = 1.99, *p* = 0.048). The subjects who exercised five times or more per week showed the higher degree of performance of exercise behavior than the subjects who exercised four times or less per week (t = −7.19, *p* < 0.001) while the subjects who exercised for more than one hour each time showed the higher degree of performance of exercise behavior than the subjects who exercised for less than one hour each time (t = −3.47, *p* < 0.001).

### 4.3. Degree of Information for Exercise, Motivation for Exercise, Sport Commitment, Perceived Benefits of Exercise, Perceived Barriers of Exercise, Exercise-Related Social Support, Exercise Self-Efficacy, and Performance of Exercise Behavior

In the mean score, the information for exercise of middle-aged men was 0.84 ± 0.24 out of 1 point and the motivation for exercise was 5.34 ± 0.99 out of 7 points. The degree of sport commitment was 3.79 ± 0.55 out of 5 points; the perceived benefits of exercise was 4.09 ± 0.58 out of 5 points; the perceived barriers to exercise was 2.57 ± 0.77 out of 5 points; and the exercise-related social support was 3.82 ± 0.67 out of 5 points. The exercise self-efficacy was 6.99 ± 1.58 out of 10 points and the degree of performance of exercise behavior was 3.04 ± 0.58 out of 4 points (Table 2).

### 4.4. Relationships of Information for Exercise, Motivation for Exercise, Sport Commitment, Perceived Benefits of Exercise, Perceived Barriers of Exercise, Exercise-Related Social Support, Exercise Self-Efficacy, and Performance of Exercise Behavior

Table 3 is the results of correlations between performance of exercise behavior and variables of middle-aged men. The performance of exercise behavior showed correlations with information for exercise (r = 0.20, *p = 0.008*), motivation for exercise (r = 0.63, *p* < 0.001), sport commitment (r = 0.60, *p* < 0.001), perceived benefits of exercise (r = 0.64, *p* < 0.001), perceived barriers to exercise (r = −0.62, *p* < 0.001), exercise-related social support (r = 0.51, *p* < 0.001), and exercise self-efficacy (r = 0.94, *p* < 0.001) in the statistically significant level. In other words, when the information for exercise, motivation for exercise, sport commitment, perceived benefits of exercise, exercise-related social support, and exercise self-efficacy were higher, and when the perceived barriers to exercise was lower, the performance of exercise behavior of middle-aged men got higher.

Additionally, all the information for exercise, motivation for exercise, sport commitment, perceived benefits of exercise, exercise-related social support, and exercise self-efficacy showed positive correlations with each other in the statistically significant level while the perceived barriers to exercise showed negative correlations with all the variables.

### 4.5. Factors Affecting the Performance of Exercise Behavior of Middle-Aged Men

Table 4 is the results of analyzing the factors affecting the performance of exercise behavior of middle-aged men belonging to exercise clubs. In order to verify the effects of information for exercise, motivation for exercise, perceived benefits of exercise, perceived barriers to exercise, exercise-related social support, and exercise self-efficacy on the performance of exercise behavior, this study verified the basic hypothesis of regression analysis before conducting the hierarchical regression analysis.

In the results of verifying the hypothesis of regression analysis, the result of Durbin-Watson test for verifying the independence was 1.84 close to 2. As there was no autocorrelation between model error terms, it satisfied the normality distribution hypothesis of residual. In the results of examining the P-P chart for verifying the independence for verifying the normality of error term, a normal distribution was shown. All the tolerance limits were in 0.25–0.90, and the variance inflation factor was in 1.10–3.12 which was not over the reference value (10), so there were no problems with multicollinearity.

In the step 1 hierarchical analysis by controlling exogenous variables, the hierarchical regression analysis was conducted by including age, religion, economic level, health condition, smoking, drinking, the number of exercises per week, and hours of each exercise that showed significant differences in the performance of exercise behavior in the univariate analysis. The variables were treated as dummy variables. In model 1, the health condition (β = 0.33, *p* < 0.001), smoking (β = −0.16, *p* = 0.016), the number of exercises per week (β = 0.28, *p* < 0.001), and hours of each exercise (β = 0.05, *p* = 0.041) had significant effects, which explained 38.4% of the performance of exercise behavior of total middle-aged men (F = 12.79, *p* < 0.001).

In the results of putting the main variables required for changes in exercise behavior such as information for exercise, motivation for exercise, sport commitment, perceived benefits of exercise, perceived barriers to exercise, and exercise-related social support in the step 2 analysis, it was more additionally explained by 21.7%. The information for exercise (β = −0.25, *p* < 0.001), motivation for exercise (β = 0.20, *p* = 0.032), sport commitment (β = 0.17, *p* = 0.025), and perceived barriers to exercise (β = −0.29, *p* < 0.001) were verified as significant predictive variables, which explained 60.1% of the performance of exercise behavior (F = 18.05, *p* < 0.001).

In the results of putting the main variable of exercise behavioral skills such as exercise self-efficacy in the step 3 analysis, it was more additionally explained by 2.9%. The exercise self-efficacy (β = 0.30, *p* < 0.001) was verified as a significant predictive variable, which explained 63.0% of the performance of exercise behavior (F = 19.10, *p* < 0.001).

## 5. Discussions

This study was performed to provide the basic data for establishing the measures for helping middle-aged men’s performance of exercise behavior by establishing the IMBR model through Information–Motivation–Behavioral skill model and literature review and verifying the factors affecting middle-aged men’s performance of exercise behavior.

As the research subjects of this study, the middle-aged men showed the relatively high degree of performance of exercise behavior (3.04 out of 4 points). Even though it was difficult to directly compare as there were no research that used the same instrument targeting the middle-aged men, in the research by Lee [22] using the same exercise behavior instrument targeting the ordinary middle-aged women instead of exercise club members, the score was 1.92. Even in the research by Oh [17], it was 2.0, which was a bit lower than the results of this study. It must be because the middle-aged men of this study were members of exercise clubs who preferred exercise based on greater interest and practice in exercise than ordinary people. The reason why the middle-aged men were targeted was that they were the subjects who should perform exercise behavior as physical activity to prevent the risk of metabolic syndrome as they were highly possible to be exposed to chronic diseases compared to middle-aged women, and there was a relatively clear correlation between physical activity and metabolic syndrome [37]. Thus, as the subjects [7] who would need it, the people who were basically performing exercise were selected. The lack of exercise is the main cause for the declined physical strength and chronic diseases, and the performance of exercise behavior has effects on the improvement of health, body functions, and life quality [1,38], so the communities and industries would need to encourage physical activities and organization of exercise clubs in order to improve health by habituating exercise behavior.

The middle-aged men’s performance of exercise behavior showed positive correlations with information for exercise, motivation for exercise, sport commitment, perceived benefits of exercise, exercise-related social support, and exercise self-efficacy while the perceived barriers to exercise showed negative correlations with them. The results of this study were supported by the results of the research [5] showing positive correlations of exercise behavior adherence intention, motivation for participation, and sport commitment of middle-aged men in exercise clubs, and showing the high intention to perform exercise behavior when the motivation for participation and sport commitment were higher, and also another research [13] showing a positive correlation between autonomous motivation and changes in exercise behavior of adult male workers. As individuals’ knowledge about behavioral change, the information is directly related to the performance of health behavior [6]. The Ministry of Health and Welfare of Korea is distributing the exercise guidelines through video and booklet [39], and the local governments are providing information for exercise and inducing practice by providing administrative and financial support through sports centers. Because the information for exercise is related to exercise behavior, the information should be continuously provided for the performance of exercise behavior.

Additionally, the motivation for exercise has a positive correlation with the performance of exercise behavior of middle-aged men, so the motivation for exercise that makes them choose exercise and maintain the exercise behavior promotes exercise behavior. The motivation for participation in foot volleyball club raised the continuous participation intention [5], and when the motivation for participation was higher in participants in sports for all, they could feel a sense of accomplishment based on fun, pride, and vitality of exercise [11]. When the motivation for exercise was higher in middle-aged women, the degree of performance of exercise behavior was high [40], which supported the results of this study. The motive by the IMB model is another element that decides the performance of health-related behavior [6], so the motivation is an important element that makes physical activity [7]. As a preparation process of choosing a behavior, it becomes the source of energy to lead goals [41]. Because the lack of motivation was an important reason for dropping out in actual exercise program [42], it would be necessary to encourage the motivation for exercise, so the middle-aged men can continuously perform exercise behavior.

When the degree of sport commitment of middle-aged men was higher, the degree of performance of exercise behavior was high. As the sport commitment means the desire, resolution, and attachment to continuously participate in exercise, it has effects on the continuous participation in exercise behavior [28]. When the degree of sport commitment of bowling participants was higher, their continuous participation and life satisfaction got higher [12], which supported the results of this study. In both Eastern and Western worlds, the sport commitment makes people naturally perform exercise behavior based on the feeling addicted to exercise, strengthens their desire to continue it [24], and brings about positive state and pleasure [43].

The perceived benefits of exercise, perceived barriers to exercise, and exercise-related social support of middle-aged men showed high correlations. Thus, when the benefits of exercise were perceived as high, when the barriers to exercise were perceived as low, and when the exercise-related social support was higher, the degree of performance of exercise behavior was high. The perceived benefits, perceived barriers, and social support are variables revealed in Pender’s health promotion model [44]. In the results of researching this in connection with exercise behavior, the performance of exercise behavior was related to the perceived benefits of exercise [20,22,45], perceived barriers to exercise [20,22,46], and social support [20,46], which supported this study. It would be necessary to adjust the environment and situation of middle-aged men, so that they could perceive the benefits of exercise as high, the barriers as low, and that they receive the material, informative, emotional, and social support in relation to exercise.

Additionally, when the exercise self-efficacy was higher, the performance of exercise behavior was high. In the Information–Motivation–Behavioral skill model, the information for exercise and motivation for exercise raised exercise behavior by going through the exercise behavioral skill such as exercise self-efficacy [6]. In the research targeting the middle-aged men [15], male workers [47], middle-aged women [20], college students [46], and patients with heart failure [7], the exercise self-efficacy or self-efficacy showed high correlations with health promotion behaviors such as exercise behavior, which supported the results of this study. Self-efficacy is an ability to organize and perform behavior well [44]. As the health promotion behavior was increased through behavioral skills in the Information–Motivation–Behavioral skill model, the exercise self-efficacy which is the subjective belief in the ability to perform behavioral change had positive effects on exercise behavior [6]. Thus, regardless of gender, age, and matter of disease of subjects, the self-efficacy is a main variable related to behavior to every subject, so the measures for increasing self-efficacy of middle-aged men should be sought for.

In the results of conducting the hierarchical regression analysis on the factors affecting the middle-aged men’s performance of exercise behavior in step 1, the health condition, smoking, the number of exercises per week, and hours of each exercise time had effects on it. The health condition of male office workers was the main factor affecting the health promotion behavior [47]. When college students perceived their own health condition as greater, they habitually performed exercise. Additionally, when they perceived their own health as greater, they could perform lots of health promotion behaviors [48]. The great perception of health condition could become the source of energy to regularly and habitually perform exercise. In the Information–Motivation–Behavioral skill model, the strong health condition and personal characteristics have effects on the model, which is directly and powerfully related to compliant behavior [6].

Smoking can have negative effects on regular exercise as the harmful substances of cigarettes can cause hypoxia in body [49]. For this reason, the subjects of this study seem not to smoke for exercise behavior they desire. Compared to middle-aged men who smoke, the middle-aged men who exercise without smoking can show more advisable results of BMI, blood pressure, blood sugar, and total cholesterol, so it is important to keep the habit of staying away from smoking. Additionally, there were differences in health promotion behavior of male office workers according to the number of exercise per week and hours of each exercise [47], so the exercise-related factors could be the main factors affecting the performance of exercise behavior. Thus, in the intervention for middle-aged men’s performance of exercise behavior, it would be necessary to consider the health condition and smoking in the general characteristics, and the number of exercise and hours of each exercise in the exercise-related characteristics.

In the step 2, among the information for exercise and motivation for exercise of Information–Motivation–Behavioral skill model, as the factors required for behavioral change, and the factors revealed in preceding researches, the sport commitment and perceived barriers to exercise were shown as the main factors affecting the middle-aged men’s performance of exercise behavior. In the Information–Motivation–Behavioral skill model, providing effective information and motivation for action to individuals could bring about positive health results. Information promotes health behavior by providing knowledge about exercise while the motive is individuals’ perception of attitude, belief, and social norms. When individuals’ attitude toward exercise behavior is greater, and when the social support for exercise behavior is highly perceived, the compliance is increasing, which is led to the high possibility of performing health behaviors such as exercise [6]. Thus, the information and motivation could become main factors affecting the exercise behavior.

Additionally, in the results of reviewing domestic/foreign literature on the factors deciding the exercise behavior, among the health promotion models mostly used as the conceptual framework of research on exercise, the factors such as barriers and benefits were verified as useful variables for health promotion [21,38.45.46]. Additionally, as the basic psychological structure that provides pleasure by raising the desire for continuous participation in exercise, the sport commitment enables middle-aged men to continue their exercise behavior by increasing their intention to continuously participate in exercise [5]. Thus, when they are provided with information for exercise and motivated for exercise [7,40,41], when they commit more to exercise [41], and when their barriers to exercise are reduced [21,46,47], they can perform exercise behavior well, and the quality of their lives could be improved. For middle-aged men’s performance of exercise behavior, the variables such as information for exercise, motivation for exercise, sport commitment, and perceived barriers to exercise should be considered.

In the results of analysis including the exercise self-efficacy of exercise behavioral skills in the step 3, the information for exercise, motivation for exercise, perceived barriers to exercise, and exercise self-efficacy of middle-aged men had effects on the performance of exercise behavior of middle-aged men, and the explanatory power of those four variables was 63.0%. In case of including the general characteristics in the step 1, it explained 38.4% of exercise behavior. In case of adding the factors required for exercise behavior in the step 2, the explanatory power was raised to 60.1%. In case of including the exercise self-efficacy of exercise behavioral skills in the step 3, the explanatory power was increased to 63.0%. According to the explanation of the Information–Motivation–Behavioral skill model, the information and motivation for exercise were not good enough to complete the exercise behavior. When the efficacy of exercise behavioral skills was added, the behavioral change could be completed [6,7]. The information for exercise and motivation for exercise had effects on exercise self-efficacy, and the exercise self-efficacy had effects on the performance of exercise behavior [7].

In the results of providing information and motivation in research applying the IMB model, the rational medication behavior [50] and self-management behavior [51] of patients were greatly improved, and especially, the self-efficacy of behavioral skills was a main predictive factor of medication behavior and exercise self-management behavior. Through the exercise self-efficacy of behavioral skills, the advisable behaviors such as exercise behavior were performed. Additionally, when the score of exercise self-efficacy was rising by one point in adults residing in community, the probability of performing exercise was increased by 1.99 times [15], which showed the behavioral skills such as self-efficacy would be an important predictive factor of performance of exercise behavior. Therefore, the intervention for the improvement of exercise self-efficacy should be developed.

As the research subjects, through the convenience sampling method, this study selected the members of exercise clubs who were exercising, so it would be limited to generalize the results of this study. It is proposed to conduct a follow-up study on middle-aged men who are not members of the sports club. Because the general characteristics and exercise-related characteristics such as health condition, smoking, the number of exercises per week, and hours of each exercise were revealed as the factors affecting the performance of exercise behavior, it would be necessary to design a research considering this. The strength of this study is to reveal the factors affecting the performance of exercise behavior by expanding the existing Information–Motivation–Behavioral skill model targeting the middle-aged men.

## 6. Conclusions

The factors affecting the middle-aged men’s performance of exercise behavior were health condition, smoking, the number of exercises per week, and hours of each exercise in the step 1. In the results of including the factors required for changes in exercise behavior in the step 2, the information for exercise, motivation for exercise, sport commitment, and perceived barriers to exercise were the factors affecting the performance of exercise behavior. In the results of including the variable of exercise behavioral skills in the step 3, the information for exercise, motivation for exercise, perceived barriers to exercise, and exercise self-efficacy were the factors affecting the performance of exercise behavior. This is the result of expanding the theory by adding (IMBR model, Figure 2) the factors required for changes in exercise behavior to Fisher’s Information–Motivation–Behavioral skill model, Therefore, this study could provide the basic data of exercise behavior performance intervention program for middle-aged men utilizing the factors affecting the performance of exercise behavior revealed in this study, which should be helpful for improving the performance of exercise.

## Figures and Tables

**Figure 1 healthcare-11-00100-f001:**
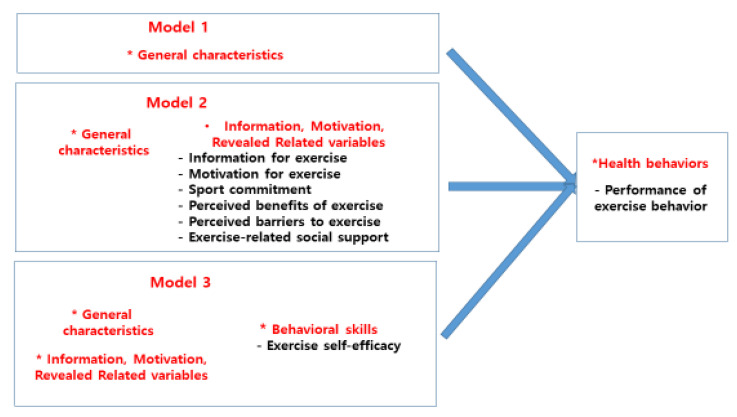
Conceptual framework for this study. * equals spot. There is no special meaning.

**Figure 2 healthcare-11-00100-f002:**
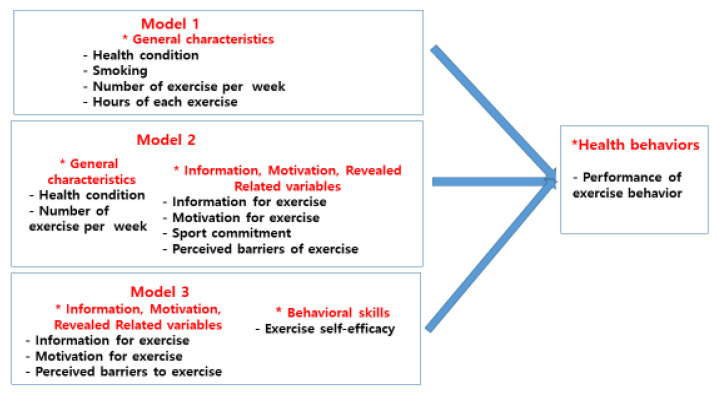
Information–Motivation–Behavioral skill-Revealed related variables (IMBR) Model. * equals spot. There’s no special meaning.

**Table 1 healthcare-11-00100-t001:** Differences in the performance of exercise behavior according to the general characteristics (*N* = 171).

Variables	Classification	*n*	%	Mean	SD	t/F	*p*-Value Scheffe Test
Age (year)	41–49	113	66.1	2.97	0.59	−2.09	0.038
	50–59	58	33.9	3.17	0.53		
Religion	Yes	62	36.3	2.92	0.56	2.09	0.038
	No	109	63.7	3.11	0.58		
Marital state	Married	158	92.4	3.05	0.58	0.45	0.584
	Unmarried	13	7.6	2.96	0.58		
Economic level	High ^a^	18	10.5	3.44	0.43	7.53	0.001
	Middle ^b^	130	76.0	3.03	0.54		a > b, c
	Low ^c^	23	13.5	2.76	0.74		
Education	Graduated from high school or lower	25	14.6	2.96	0.60	−0.75	0.452
	Graduated from college or higher	146	85.4	3.05	0.58		
Job	Office work	102	59.6	3.02	0.61	−0.66	0.510
	Production work or self-employed	69	40.4	3.08	0.54		
Health condition	Bad	56	32.7	2.62	0.54	−7.67	<0.001
	Good	115	67.3	3.25	0.48		
Smoking	Yes	39	22.8	2.70	0.55	4.35	<0.001
	No	132	77.2	3.14	0.55		
Drinking	Yes	105	60.1	2.97	0.59	1.99	0.048
	No	66	39.9	3.15	0.55		
The number of diseases	1	165	96.5	3.05	0.59	0.89	0.374
	≥2	6	3.5	2.83	0.37		
Number of exercise per week	≤4	127	74.3	2.88	0.35	−7.19	<0.001
	≥5	44	25.7	3.52	0.60		
Hours of each exercise	Less than one hour	123	71.9	2.95	0.56	−3.47	0.001
	More than one hour	48	28.1	3.28	0.57		

^a^, ^b^, ^c^ is expressed so that the meaning of the Scheffe test at the right end can be known.

**Table 2 healthcare-11-00100-t002:** Degree of research variables (*N* = 171).

Variables	Mean	SD	Range
Information for exercise	0.84	0.24	0.14–1
Motivation for exercise	5.34	0.99	2.92–7
Sport commitment	3.79	0.55	1.42–5
Perceived benefits of exercise	4.09	0.58	2.72–5
Perceived barriers to exercise	2.57	0.77	1–4.30
Exercise-related social support	3.82	0.67	1.71–5
Exercise self-efficacy	6.99	1.58	2.08–10
Performance of exercise behavior	3.04	0.58	1.14–4

**Table 3 healthcare-11-00100-t003:** Relationships of information for exercise, motivation for exercise, sport commitment, perceived benefits of exercise, perceived barriers of exercise, exercise-related social support, exercise self-efficacy, and performance of exercise behavior.

Variables	Information for Exercise r (*p*)	Motivation for Exercise r (*p*)	Sport Commitment r (*p*)	Perceived Benefits of Exercise r (*p*)	Perceived Barriers to Exercise r (*p*)	Exercise-Related Social Support r (*p*)	Exercise Self-Efficacy r (*p*)	Performance of Exercise Behavior r (*p*)
Information for exercise	1							
Motivation for exercise	0.49 (<0.001)	1						
Sport commitment	0.27 (<0.001)	0.65 (<0.001)	1					
Perceived benefits of exercise	0.49 (<0.001)	0.76 (<0.001)	0.69 (<0.001)	1				
Perceived barriers to exercise	−0.49 (<0.001)	−0.66 (<0.001)	−0.41 (<0.001)	−0.66 (<0.001)	1			
Exercise-related social support	0.28 (<0.001)	0.50 (<0.001)	0.57 (<0.001)	0.58 (<0.001)	−0.37 (<0.001)	1		
Exercise self-efficacy	0.36 (<0.001)	0.61 (<0.001)	0.60 (<0.001)	0.72 (<0.001)	−0.67 (<0.001)	0.51 (<0.001)	1	
Performance of Exercise behavior	0.20 (0.008)	0.63 (<0.001)	0.60 (<0.001)	0.64 (<0.001)	−0.62 (<0.001)	0.51 (<0.001)	0.69 (<0.001)	1

**Table 4 healthcare-11-00100-t004:** Factors affecting the performance of exercise behavior of middle-aged men.

Variables	Model 1					Model 2					Model 3				
	B	SE	β	t	*p*	B	SE	β	t	*p*	B	SE	β	t	*p*
Age	0.13	0.08	0.10	1.67	0.097	0.04	0.06	0.04	0.70	0.482	0.03	0.06	0.03	0.57	0.568
Religion (yes)	−0.08	0.07	−0.07	−1.10	0.275	−0.02	0.06	−0.02	−0.37	0.710	−0.45	0.06	−0.04	−0.77	0.444
Economic status (high)	0.20	0.15	0.11	1.31	0.194	0.06	0.13	0.03	0.46	0.643	0.06	0.12	0.03	0.52	0.607
Economic status (middle)	0.01	0.11	0.01	0.09	0.927	−0.01	0.09	<0.001	−0.01	0.995	0.04	0.08	0.03	0.50	0.618
Health condition (good) (ref = bad)	0.40	0.09	0.33	4.73	<0.001	0.14	0.07	0.11	1.90	0.060	0.14	0.07	0.12	1.98	0.050
Smoking (no) (ref = yes)	−0.22	0.09	−0.16	−2.43	0.016	−0.10	0.07	−0.07	−1.29	0.201	−0.11	0.07	−0.08	−1.57	0.118
Drinking (no) (ref = yes)	−0.02	0.08	−0.01	−0.22	0.823	−0.08	0.06	−0.07	−1.22	0.226	−0.11	0.06	−0.09	−1.71	0.089
Number of exercise per week (≥5) (ref = ≦4)	0.36	0.09	0.28	3.94	<0.001	0.18	0.08	0.13	2.24	0.027	0.12	0.08	0.09	1.52	0.131
Hours of each exercise (More than one hour) (ref = less than one hour)	0.07	0.08	0.05	0.83	0.041	−0.45	0.07	−0.04	−0.67	0.507	−0.09	0.07	−0.07	−1.34	0.183
Information for exercise						−0.62	0.15	−0.25	−4.21	<0.001	−0.60	0.14	−0.25	−4.25	<0.001
Motivation for exercise						0.11	0.05	0.20	2.17	0.032	0.13	0.05	0.22	2.47	0.015
Sport commitment						0.18	0.08	0.17	2.27	0.025	0.13	0.08	0.12	1.64	0.102
Perceived benefits of Exercise						0.12	0.09	0.12	1.33	0.186	0.03	0.09	0.03	0.31	0.755
Perceived barriers of Exercise						−0.22	0.06	−0.29	−3.85	<0.001	−0.15	0.06	−0.19	−2.54	0.012
Exercise-related social support						0.07	0.06	0.08	1.13	0.259	0.04	0.06	0.04	0.64	0.521
Exercise self-efficacy											0.11	0.03	0.30	3.65	<0.001
R^2^	0.417					0.636					0.665				
Adjusted R^2^	0.384					0.601					0.630				
Δ Adjusted R^2^ (*p*)						0.217 (<0.001)					0.029 (<0.001)				
F *(p)*	12.79 (<0.001)					18.05 (<0.001)					19.10 (<0.001)				

SE = standard error; ref = reference.

## Data Availability

The data underlying this article will be shared upon reasonable request from the corresponding author.

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
