# Peer review of "Factors Influencing Performance of Exercise Behavior of Middle-Aged Men with Chronic Disease Based on the Information–Motivation–Behavioral Skill-Revealed-Related Variables (IMBR) Model"

_healthcare, 2022, doi:10.3390/healthcare11010100_

Round 1
Reviewer 1 Report
The manuscript is well-written with strong design and data analysis, with only minor comments for wording clarity:
Line 32: "among the OECD countries..." Please list words of the acronym
Line 35: "in health maintenance/enhancement..." Please separate out the word in the slashes for readability, probably should read as maintenance and enhancement. Same for Line 41, 43, 72, 121, 230
The acronyms IMB and IMBR seem to be used interchangeably in the title and sporadically throughout the document, please choose one use and instead write the other one out, to avoid confusion.
Lines 168-169: "The significance of participating in the study should be raised by giving feedbacks on the results afterwards." Incomplete sentence or is grammatically incorrect, please check.
Author Response
|
Reviewer 1 |
|
|
Comments and SuggestionsThe manuscript is well-written with strong design and data analysis, with only minor comments for wording clarity: |
Thank you very much for your review carefully so that it can be a good paper |
|
Line 32: "among the OECD countries..." Please list words of the acronym |
revised it Organization for Economic Cooperation and Development (OECD) |
|
Line 35: "in health maintenance/enhancement..." Please separate out the word in the slashes for readability, probably should read as maintenance and enhancement. Same for Line 41, 43, 72, 121, 230 |
separate out the word as follows Line 35: maintenance and enhancement Line 41: maintain and improve Line 43: mental and physical health Line 72: behavioral change and maintenance Line 121: cognitive and emotional variables Line 230: modified |
|
The acronyms IMB and IMBR seem to be used interchangeably in the title and sporadically throughout the document, please choose one use and instead write the other one out, to avoid confusion. |
IMB was deleted, Information-Motivation-Behavioral skill model was written, and IMBR was used as an abbreviation. |
|
Lines 168-169: "The significance of participating in the study should be raised by giving feedbacks on the results afterwards." Incomplete sentence or is grammatically incorrect, please check. |
Changed the sentence. In the future, the results will be feedback to the subjects to increase the meaning and significance of participation in the study. |

Reviewer 2 Report
This study analyzed the factors influencing performance of exercise behavior of middle-aged men with chronic disease by adding variables through literature based on IMB model,which is meaningful. However, in the paper,three are too many problems listed belowed,so please check them.1、The content of line between 29 and32 is lackng of theoretical referrence.
2、The appropriateness of grammar and vocabulary needs improving greatly.
3、The content of line between 32 and 34 is lackng of theoretical referrence.
4、In line 111, the conception of “sport commitment” needs to be added.
5、In the proportion of “introduction” , the hypotheses of sport commitment, perceived benefits of exercise and perceived barriers to exercise lake adequate proof.
6、The sample caculated by G-power is 179,while this only distributed 179 questionnaires,which did not consider the collection rate of the questionnaire.
7、In line 154,what does “154,“they are exercising in daily life based on their ability to communicate.” mean?
8、In lines between 178 and 179,what does“the researchers made a tool of knowledge 178 about exercise by testing the validity of two exercise experts and a professor focusing on 179 the literature by Kim et al.” mean?
9、in line between 197 and198,“In the research by Jin [27], the Cronbach’s ⍺ for the reliability of personal motivation 197 was 0.90 while the reliability of social motivation was 0.90.”,the questionnaire has been revised, so it is not necessary to introduce the reliability of former questionnair. The same problem exists in line 205,218,and 237.
10、In line 214, “This study used the tool modified/complemented by Oh [20]”,Oh?
11、In figure 4, the abbreviation of acronym should be noted.
12、The content in Figure 1 doesn't match the model of hierarchical multiple regression analysis.
Author Response
|
Reviewer 2 |
|
|
This study analyzed the factors influencing performance of exercise behavior of middle-aged men with chronic disease by adding variables through literature based on IMB model,which is meaningful. However, in the paper,three are too many problems listed belowed,so please check them. 1.The content of line between 29 and32 is lackng of theoretical referrence |
Thank you very much for your careful review. I worked hard to revise what you pointed out.
I added reference number one. |
|
2.The appropriateness of grammar and vocabulary needs improving greatly. |
This paper is the result of commissioning to a specialized editing company. The paper has been modified. |
|
3.The content of line between 32 and 34 is lackng of theoretical referrence. |
I added reference number one. |
|
4.In line 111, the conception of “sport commitment” needs to be added.
|
Add the definition of Sport commitment in line 112-113. Sport commitment refers to the optimal psychological state that occurs when you are completely immersed in exercise activities [12]. |
|
5.In the proportion of “introduction” , the hypotheses of sport commitment, perceived benefits of exercise and perceived barriers to exercise lake adequate proof. |
Add it in line 117-118 The higher the degree of sport commitment of middle-aged men, the higher the intention to continue participation [5] and the higher the performance of exercise behavior. Add it in line 125-127 The higher the benefit of exercise and the lower the disability of exercise, the higher the ability to perform exercise [22]. Middle-aged men will recognize the benefit of exercise and the lower the disability of exercise, the better they will perform exercise behavior.
|
|
6.The sample caculated by G-power is 179,while this only distributed 179 questionnaires,which did not consider the collection rate of the questionnaire. |
The number of people calculated in consideration of the independent variable related to the dependent variable, the statistical processing of regression analysis, and the effective size was 171, and 179 questionnaires were distributed in consideration of the dropout rate of 5%. All of them were collected, but 171 questionnaires were included in the study, except for 8 questionnaires that were not appropriate. In line 158-160, add it. Considering the 5% dropout rate, including the number of unavailable questionnaires, such as collection difficulties and unavailable questionnaires, |
|
7、In line 154,what does “154,“they are exercising in daily life based on their ability to communicate.” mean? |
Changed it in line 164-165 ~~~ exercising in their daily lives, understands the purpose of research and how to collect data, and allowed to participate in the study. |
|
8、In lines between 178 and 179,what does “the researchers made a tool of knowledge 178 about exercise by testing the validity of two exercise experts and a professor focusing on 179 the literature by Kim et al.” mean? |
Changed the sentence. The researchers completed the instrument by referring to the literature of Kim et al. [2] and the contents of information for exercise presented in Song's [25] study. After that, two exercise experts and one professor of physical education checked the validity of the instrument. |
|
9、in line between 197 and198,“In the research by Jin [27], the Cronbach’s ⍺ for the reliability of personal motivation 197 was 0.90 while the reliability of social motivation was 0.90.”,the questionnaire has been revised, so it is not necessary to introduce the reliability of former questionnair. The same problem exists in line 205,218,and 237. |
I deleted the previous reliability of all tools and corrected the sentence. |
|
10、In line 214, “This study used the tool modified/complemented by Oh [20]”,Oh? |
Revised it This study used instrument that Oh [20] modified instrument developed by Lee [22] based on instrument by Sebrist et al., [29] and instrument by Stenhardt, Dishman [30]. |
|
11、In figure 4, the abbreviation of acronym should be noted. |
Revised, SE=standard error |
|
12、The content in Figure 1 doesn't match the model of hierarchical multiple regression analysis.
|
Figure 1 and Figure 2 are redrawn to match the hierarchical regression analysis. Thank you. Figure1: in line 149 Figure2: in line 595 |

Reviewer 3 Report
This is an important topic given that physical activity and exercise plays an big part in preventing chronic diseases such as obesity, high blood pressure and diabetes. The information motivational behavioral skill model is introduced and extended to add variables affecting the performance of exercise behavior.
The introduction, methodology, results, analysis and discussion are well presented.
It is recognised that the study population is not representative of all middle aged men as the participants are selected from members of an exercise club.
It will be good to have a follow up study including middle aged men who are not members of exercise clubs. Being a member of an exercise club in itself can be a motivational factor.
Introduction This is an interesting article that will add to the literature on physical exercise in middle aged men and the prevention of chronic diseases such as hypertension, obesity and diabetes. The introduction is supported by citing the existing literature and the Information-Motivation-Behavioral skill model (IMB model). The paper goes further to investigate how additional variables affect performance of exercise behavior such as, sports commitment, perceived benefits, perceived barriers and exercise related social support. Methodology The study has been approved by an ethics committee. Participants are middle aged males who are members of a sports club. Inclusion and exclusion criteria stated. The participants give their consent and know that they can withdraw from the study at any time. All study information is stored safely. The methodology is clearly explained and the study could be replicated. The authors recognise and state their limitations. Results and analysis The results are clearly presented. Appropriate analysis methods used and presented clearly. . General comments The study participants do not represent the general population so the results can not be generalised. Given that the study participants belong to a sports club they are more likely to have sports commitment, more information on the benefits of exercise and social support.. The information is collected using a self administered questionnaire. The perceptions of individuals can change and are dependent on many other factors influencing the participants. There is a need to recruit participants who are not members of sports clubs but exercise independently and represent the wider population to be able to make generalised statements. The study expands on the IMB model and thus creates a new model, the IMBR model. This adds to the literature.
This study should be published as it is good for an initial study to add to the existing literature, which will be followed up on. Follow up studies can have participants that represent the general population and explore the different ways of informing individuals of the benefits of sports and the effect this will have on exercise self efficiency.
Author Response
|
Reviewer 3 |
|
|
This is an important topic given that physical activity and exercise plays an big part in preventing chronic diseases such as obesity, high blood pressure and diabetes. The information motivational behavioral skill model is introduced and extended to add variables affecting the performance of exercise behavior. The introduction, methodology, results, analysis and discussion are well presented. It is recognised that the study population is not representative of all middle aged men as the participants are selected from members of an exercise club. |
Thank you very much for understanding the research purpose and effort of the researchers. |
|
It will be good to have a follow up study including middle aged men who are not members of exercise clubs. Being a member of an exercise club in itself can be a motivational factor. |
Thank you so much. Add the sentence Line 544-545: It is proposed to conduct a follow-up study on middle-aged men who are not members of the sports club. |

Round 2
Reviewer 2 Report
After therevision, the paper has been improved greatly,congratulations!
However,the lines between 107 and 116 were not presented logically,so please check.
Author Response
In line 107-116, we corrected it as you pointed out.
I modified IMB to Information-Motivation-Behavioral skill model and modified physical to exercise which can be confusing.
Thank you.
